# Accuracy of Catheter Positioning during Left Subclavian Venous Access: A Randomized Comparison between Radiological and Topographical Landmarks

**DOI:** 10.3390/jcm11133692

**Published:** 2022-06-27

**Authors:** Sun Key Kim, Jung Hwan Ahn, Yoon Kyung Lee, Bo Young Hwang, Min Kyung Lee, Il Seok Kim

**Affiliations:** 1Department of Anesthesiology and Pain Medicine, Kangdong Sacred Heart Hospital, Hallym University College of Medicine, Seoul 05355, Korea; ksk8325@naver.com (S.K.K.); ykleeanes@gmail.com (Y.K.L.); medi1@hanmail.net (B.Y.H.); sb580623@naver.com (M.K.L.); 2Department of Emergency Medicine, Ajou University School of Medicine, Suwon 16499, Korea; erdrajh@naver.com

**Keywords:** central venous catheters, echocardiography, left subclavian vein, superior vena cava, ultrasound

## Abstract

Left subclavian venous access increases the risk of vascular damage and thrombosis based on the catheter course and location of the catheter tip. We investigated the accuracy of tip positioning with conventional landmarks using transesophageal echocardiography. The carina as a radiological landmark and the right third intercostal space as a topographical landmark were selected for tip positioning within the target zone, defined as 2 cm above and 1 cm below the right atrial junction. A total of 120 participants were randomized into two groups. The catheter insertion depth was determined as 1.5 cm more than the distance between the venous insertion point and the carina via the right first intercostal space in the radiological group, and between the venous insertion point and the right third intercostal space via the right first intercostal space in the topographical group. The determined insertion depth and actual distance to the right atrial junction of the radiological and topographical groups were 19.5 cm and 20.5 cm, and 19.8 cm and 20.4 cm, respectively. Acceptable positioning was more frequent in the topographical group (96.4% vs. 85.7%; *p* = 0.047). The catheter tip is more accurately positioned in the distal superior vena cava using topographical landmarks than radiological landmarks.

## 1. Introduction

The safety and function of central venous catheter positioning based on site selection remains debatable [1,2]. It is recommended that the catheter tip should be placed in the superior vena cava (SVC) above the pericardial sac to prevent serious complications such as cardiac tamponade [3].

During left subclavian venous access, the catheter tip may be positioned in the middle portion of the innominate vein to ensure a parallel course and prevent SVC abutment. However, in this proximal position, it is prone to thrombosis due to the relatively small venous diameter, malfunction of the catheter owing to extravasation of the proximal access site, and infection in the case of repositioning [4,5]. In the middle position, including the upper and middle SVC, the catheter may result in vascular irritation due to abutment with the SVC at a steep angle [6,7]. In contrast, when positioned in the distal SVC, close to the right atrial junction, it reduces the risk of vascular damage and thrombotic complications due to the parallel pathway of the catheter tip and the large conduit of the vein [8]. Therefore, the catheter tip in this specific distal position is better suited for left subclavian venous access [9].

The conventional simple formula based on the patient’s height is not accurate for catheter tip positioning during ultrasound-guided cannulation. For right-sided venous access, the catheter tip at 1.5 cm near the carina on chest radiography would be positioned in the distal SVC close to the right atrial junction [8]. However, there is no definite landmark for catheter tip positioning during left subclavian venous access. Therefore, we planned catheter tip positioning at the distal SVC close to the right atrial junction using anatomical landmarks. The SVC is identified based on the overlying structures on coronal and axial computed tomography images. It primarily originates behind the right first intercostal space and terminates in the right atrium in the third or fourth intercostal space [10,11]. The sternal angle formed by the manubriosternal joint is easily palpable over the skin. The second costal cartilages articulate on either side of the sternal angle [12].

This study investigated the accuracy of catheter tip positioning using landmark-based methods during left subclavian venous cannulation. In this study, we determined the carina as a radiological landmark and the right third intercostal space as a topographical landmark for left subclavian venous access. The accuracy of catheter tip positioning between the two landmark-based methods was compared using transesophageal echocardiography.

## 2. Materials and Methods

### 2.1. Study Design, Ethics Statement and Study Population

This prospective randomized controlled study investigated the accuracy of catheter tip positioning by landmark-based methods during left subclavian venous cannulation. Ethical approval for this study was provided by the Institutional Review Board of Kangdong Sacred Heart Hospital in Seoul, Republic of Korea (President: Soo Young Kim, protocol number: KANGDONG 2019-03-002-001) on 26 April 2019. All the experiments were carried out in accordance with the relevant guidelines and regulations of the Declaration of Helsinki involving human subjects. All the patients signed an informed consent form prior to study enrolment. After obtaining written informed consent from each patient, we recruited 120 patients (20–80 years of age) with the American Society of Anesthesiologists physical status class 1 to 3, who were eligible for left subclavian venous cannulation before abdominal and cardiovascular surgeries between April 2019 and November 2021. The exclusion criteria were a previous history of thoracic surgery, mediastinal mass, esophageal varices, and refusal to participate. This study was registered with the Clinical Research Information Service of Korea (https://cris.nih.go.kr (accessed on 20 July 2021); identifier: KCT0006388; principal investigator: Il Seok Kim).

### 2.2. Randomization and Allocation

Participants were randomly allocated in a 1:1 ratio to either the radiological landmark group (R group) or the topographical landmark group (T group) using computer-generated randomization (www.graphpad.com/quickcalcs (accessed on 16 May 2022)). The allocation of participants was concealed in a sequentially numbered opaque envelope, and the assignment envelope was opened before cannulation.

The carina was selected as the radiological landmark using a preoperative standard erect P-A chest radiograph in suspended full inspiration in the R group. To estimate the distance between the right first intercostal space and the carina, the vertical length between the lower border of the right first costal cartilage, close to the sternum, and a horizontal line connecting the carina were measured using the picture archiving and communication system (PACS, Infinite Healthcare Co., Seoul, Korea) and an internal electronic caliper (Figure 1a). The catheter insertion depth was calculated by adding the distance between the venous insertion point and the right first intercostal space measured over the skin, the distance between the right first intercostal space and the carina on chest radiography, and an additional 1.5 cm for safety against insertion of the catheter tip in the right atrium [8].

The right third intercostal space was selected as the topographical landmark over the chest skin surface in the T group. By measuring the distance between the midpoints of the first and third intercostal spaces over the skin, the catheter insertion depth was calculated by adding the distance between the venous insertion point and the right first intercostal space and the distance between the first and third intercostal spaces (Figure 1b).

### 2.3. Procedure and Data Collection

Following general anesthesia induction, an echocardiographic probe (X7-2t transesophageal transducer; Phillips, Andover, MA, USA) was inserted into the esophagus. During cannulation, the patient was maintained in the Trendelenburg position with arms abducted. After sterile preparation and draping, the puncture site in the infraclavicular area was pre-scanned using two-dimensional ultrasonography (Affiniti 70; Phillips, Andover, MA, USA) and a high-frequency linear transducer. After palpation of the sternal angle and identifying the right first intercostal space over the skin, the distance between the venous insertion point and the midpoint of the right first intercostal space just lateral to the sternal angle, and the distance between the first and third intercostal spaces just lateral to the sternum, were measured using a sterile graduated ruler. Central venous cannulation was performed by an ultrasound-guided in-plane approach in the longitudinal view. A 20 cm long, two-lumen catheter (Arrow G^+^ard Blue Central Venous Catheter; Arrow International Inc., Reading, PA, USA) was inserted using the Seldinger technique and secured at the determined depth according to the protocol for each group. On the bicaval view of echocardiography (Figure 2), accurate positioning of the catheter tip was assessed relative to the right atrial junction, which was assumed to be at the level of the upper border of the crista terminalis [13]. We also assessed the incidence of the angle of the tip > 40° in relation to the SVC, abutment of the tip with the SVC, and flow streams hitting the vascular wall using injections of agitated saline at the radiologically or topographically predetermined insertion depth.

The actual distance between the venous insertion point and the right atrial junction was assessed using advancement or withdrawal of the catheter from the bicaval view. Following the repositioning of the catheter tip at the right atrial junction or the maximum depth of the 20 cm catheter, the catheter was fixed with a skin suture.

Postoperatively, the catheter position was rechecked using a recumbent chest radiograph upon inspiration at the bedside. Any complication related to cannulation was recorded until the removal of the catheter.

The primary outcome was the incidence of acceptable positioning of the catheter tip within the target zone, which was designated as 2 cm above and 1 cm below the right atrial junction, since this area has a large conduit of vessels and the catheter tip floats freely without impinging on the vascular wall. The secondary outcomes were the difference between the determined insertion depth and the actual distance to the right atrial junction, the incidence of the angle of the tip > 40° in relation to the SVC, tip abutment with the SVC, flow streams hitting the vascular wall, and any cannulation-related complications.

### 2.4. Statistical Analysis

#### 2.4.1. Sample Size Calculation

Based on the landmarks and calculated values for right-sided cannulation from a previous study, the sample size was calculated from the data based on our preliminary observation, in which the patients were divided into two groups with the carina and the right third intercostal space as landmarks for positioning the catheter tip within the target zone [8]. Consequently, 30 patients were included in each group, and the incidence of acceptable positioning was 83.3% (25/30) and 96.6% (29/30) in the carina and third intercostal space groups, respectively. Based on the incidence rate, an alternative hypothesis and test type were chosen as one-sided (H1: P1 < P2) and the pooled Z test, respectively. We calculated that 56 patients were required in each group to detect a difference of this magnitude with an α error of 0.05 and a desired power of 0.80, using PASS 12 (NCSS, LLC, Kaysville, UT, USA). After accounting for a dropout rate of 6%, we recruited 120 patients for this study.

#### 2.4.2. Data Analysis

Statistical analysis was performed using the SPSS version 23.0 (IBM Inc., Armonk, NY, USA). The Shapiro–Wilk test was used to assess the normal distribution of variables. Continuous variables are reported as medians (interquartile range (IQR)) and compared using the Mann–Whitney U test or the independent Student’s *t*-test, as considered appropriate. Categorical variables are presented as numbers (proportion) and compared using Fisher’s exact test or Pearson’s chi-square test, as considered appropriate. A probability value less than 0.05 was considered to be statistically significant.

## 3. Results

### 3.1. Participant Enrollment

Of the 120 patients screened for the study, six patients were excluded owing to the unavailability of echocardiography at cannulation (*n* = 2), conversion to other access sites (*n* = 3), and preoperative detection of an abnormal thoracovascular condition of persistent left SVC (*n* = 1). Accordingly, 114 participants were randomly allocated to one of the two intervention groups, with 57 participants in each group (Figure 3). All the participants underwent successful cannulation and catheter positioning, except two patients in whom the catheter tip could not be identified from the bicaval view. Misplacement of the catheter into the left internal jugular vein occurred in one patient in the R group, and an aberrant positioning of the catheter due to persistent left SVC occurred in one patient in the T group. These patients were not included in the statistical analysis. Finally, 56 patients per group were analyzed.

### 3.2. Characteristics of the Participants and Measurements

The baseline characteristics of the study participants are presented in Table 1. Sex, age, height, weight, and body mass index were comparable between the groups.

The measurements of catheter positioning are summarized in Table 2. Between the R and T groups, the determined insertion depth (19.5 [18.6–20.4] cm vs. 19.8 [18.8–20.2] cm, respectively; *p* = 0.645), the actual distance to the right atrial junction (20.5 [19.6–21.0] cm vs. 20.4 [19.5–21.0] cm, respectively; *p* = 0.802), and the difference between the measurements (0.7 [0.1–1.4] cm vs. 0.5 [0–0.8] cm, respectively; *p* = 0.171) were comparable. The proportion of acceptable positioning of the catheter tip within the target zone was higher in the T group than in the R group (96.4% vs. 85.7%, respectively, *p* = 0.047) (Figure 4). The proportion of tip positioning above the target zone was higher in the R group than in the T group (14.3% vs. 3.6%, respectively, *p* = 0.047). Tip position below the target zone was not observed in either group. The proportion of angle of the tip > 40° to the SVC, tip abutment with the SVC, and flow streams hitting the vascular wall were comparable between the groups. Until the removal of the catheter, no catheter-related complications were observed in either group.

## 4. Discussion

The principal finding of our investigation was that during left subclavian venous access, we could place the catheter tip more accurately in the distal SVC close to the right atrial junction using the topographical method.

The placement of the central venous catheter is always associated with risks, and the optimal positioning of the catheter tip is an ongoing issue, especially in left-sided venous access. Vascular injuries are possible in any position in the SVC and cardiac chamber. Most devastating complications, such as cardiac tamponade, hemothorax, and hydrothorax, were reported in case reports and were attributed to mechanical and chemical irritation to the vascular wall, which were related to parenteral delivery of hyperosmolar solutions and an acute angle from left-sided access. The most important points to be considered for preventing these events are the alignment of the catheter tip to the vessel wall, free movement of the tip, and non-impingement to the vessel wall. The distal SVC close to the right atrial junction has the advantage of ensuring a parallel pathway for the catheter tip and a large conduit during cardiac pulsation.

Several methods and landmarks are used for catheter positioning in right-sided vascular access. However, there is no definite landmark in left-sided access for catheter tip positioning in the distal SVC close to the right atrial junction.

In our study, although both the methods demonstrated comparable outcomes, the radiological method had a higher incidence of the catheter tip being located 2 cm above the right atrial junction. The catheter tip was more accurately positioned in the distal SVC close to the right atrial junction when using topographical landmarks compared to radiological landmarks.

The carina on chest radiography has been used as the landmark for catheter positioning, and when the catheter tip is positioned above the carina, it is generally accepted that the catheter tip could be located above the pericardial reflection during right-sided central venous access [14,15]. However, in left-sided access, if the catheter tip is located above the carina, it results in an acute angle with the vascular wall and increases the risk of vascular damage [7]. In catheter positioning using electrocardiogram guidance with P-wave normalization and the manubriosternal junction as a surface landmark assumed at the level of the carina, positioning the catheter tip above the carina results in a high incidence of the catheter tip being positioned at an acute angle with the vascular wall in left-sided access [16,17]. Therefore, the catheter tip would be positioned below the carina for left-sided vascular access; however, the distance for preventing intracardiac placement has not been specified [7]. The distance from the carina to the right atrial junction varies from 2.0 to 4.0 cm [18,19]. A previous study reported that the mean (standard deviation) distance from the carina to the right atrial junction was 2.6 (1.1) cm; therefore, we selected 1.5 cm as the minimum distance while positioning the catheter tip below the carina to prevent its placement in the right atrium in the R group [8]. Although the determined insertion depth, the actual distance to the right atrial junction, and the difference between the measurements were comparable between the groups, more catheter tips were positioned above the target zone (14%, 8/56), and a significant proportion of catheters were positioned below the atrial junction (21%, 12/56) in the R group. These results may be due to imprecise measurement of the right first intercostal space combined with a parallax effect on imaging of the right first rib from the central radiographic beam and the patient’s position and height, individual variability in the distance between the right first intercostal space and the carina, and underestimation of the distance from the carina to the right atrial junction as only 1.5 cm.

Based on identifiable cutaneous landmarks overlying internal structures and their respective courses, several topographical landmarks have been proposed for catheter positioning [15,20]. Using the clavicular notch on the sternoclavicular joint and the sternal angle formed by the manubriosternal joint, the catheter tip can be reliably placed in the SVC above the pericardial reflection for right-sided venous access [17]. For placing the catheter tip near the radiographic junction of the SVC and the right atrium, the right third intercostal space is a reliable surface landmark in pediatric patients [21]. One study reported that by using the lower border of the clavicular notch as the reference point for a guidewire through the brachiocephalic vein and SVC, the lower border of the right third costosternal junction was more reliable at positioning the catheter tip within 1 cm of the echocardiographic junction of the SVC and the right atrium [22]. The SVC is a confluence of the right brachiocephalic vein and left brachiocephalic vein and commonly originates at the level of the right first intercostal space on computed tomography; therefore, the right first intercostal space is a more accurate reference point than the clavicular notch based on the course of the catheter inserted in the left-sided venous access [10,11]. Therefore, for positioning the catheter tip close to the echocardiographic junction of the right atrium, we selected the right first and third intercostal spaces as landmarks to appraise the origin, course, and termination of the SVC. The results demonstrated a high proportion of catheter tips positioned within the target zone. We determined the target zone as 2 cm above and 1 cm below the right atrial junction, because this area is wide and parallel to the vascular conduit of the SVC; additionally, the catheter tip is likely to float freely without impinging on the atrial wall during cardiac contractions in echocardiographic imaging.

In the T group, the catheter tip was positioned 2 cm above the target zone in two participants. Of these, one outlier was a 31-year-old man with a height of 179 cm, and the other was a 73-year-old man with a height of 177 cm. The SVC length reportedly ranges widely from 4.4 to 10 cm on magnetic resonance imaging [18]. Additionally, in a computed tomography-based study, the termination of the SVC was more variable between the sexes and age groups in relation to the overlying surface structures than the origin of the SVC; the termination of the SVC into the right atrium was identified from the right third intercostal space to the fifth costal cartilage, and was higher in women and younger adults [23]. Therefore, the considerable variability in the SVC length and right atrial junction with the corresponding surface landmarks could have resulted in these outliers.

In the bicaval view of echocardiography, the catheter tip location could not be confirmed in two patients, who were subsequently excluded from the statistical analysis owing to incomplete follow-up. These events occurred owing to the misplacement into the internal jugular vein and the presence of a persistent left SVC. Although misplacement is less common in left-sided subclavian venous access owing to an obtuse angle of the innominate vein with the SVC, it did occur in one participant in the R group [24]. It may be associated with a tortuous path and a more distal approach from the axillary vein. Persistent left SVC is an abnormal thoracic venous condition that occurs in 0.3% of the general population [25]. This vein empties into the right atrium through the coronary sinus in up to 90% of people and is generally asymptomatic; however, it occasionally drains into the left atrium, which increases the risk of systemic embolism [26]. In our case, aberrant positioning of the catheter was discovered accidentally after cannulation. During cannulation, bubble streams in the right atrium were identified following injection of agitated saline, but the catheter tip was not detected from the bicaval view. Fortunately, this case remained uneventful after cannulation; however, awareness of the clinical implications and thoughtful examination of coronary sinus dilation using echocardiography can help avoid potential complications. These patients underwent transesophageal ultrasound for the measurement of catheter tip positioning; however, no clear benefit was observed. A confirmatory chest radiograph is still needed regardless of the calculation method used for catheter tip positioning.

There were certain limitations to our study. First, we performed cannulation solely of the left subclavian vein. Left internal jugular venous access carries a risk of vascular irritation since it exhibits two curvatures up to the right atrial junction, and achieving acceptable positioning with commercial catheters of 20 cm is challenging; therefore, we did not include the left internal jugular vein in this study. Second, we did not consider the possibility of catheter tip migration based on patients’ posture and arm movements. The participants in this study required central venous cannulation as a part of perioperative care and not permanent implantation for long-term usage. Third, we did not consider the difference in patient position and breathing pattern, wherein for the calculation using a preoperative standard P-A chest radiograph, the patient was in an upright position in full inspiration, while for the measurement using transesophageal ultrasound, the patient was in a supine position and mechanically ventilated.

In conclusion, we demonstrated that the catheter tip could be more accurately positioned in the distal SVC close to the right atrial junction using the topographical method. Therefore, we recommend using the right first and third intercostal spaces as landmarks during cannulation of the left subclavian vein for positioning of the catheter tip close to the right atrial junction. If identifying the surface landmarks is challenging, radiological landmarks, such as the carina, can be alternatively used for positioning the catheter.

## Figures and Tables

**Figure 1 jcm-11-03692-f001:**
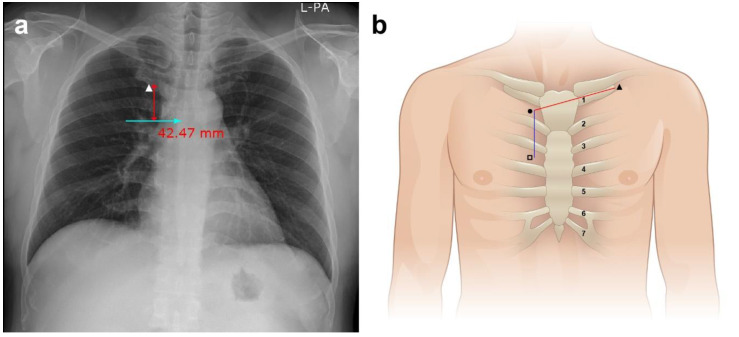
Radiological and topographical landmarks. (**a**) Chest radiograph for measuring the distance between the first intercostal space and the carina. The distance from the right first intercostal space to the carina is measured as the vertical length between the lower border of the right first costal cartilage (solid triangle) close to the sternum and a horizontal line connecting it to the carina using an electronic caliper in the radiological group. (**b**) Schematic illustration for estimating the distance from the venous insertion point through the right first intercostal space to the right third intercostal space in the topographical group. The distance is determined by adding the distance between the venous insertion point (solid triangle) and the midpoint of the right first intercostal space (solid circle) just lateral to the sternal angle, and the distance between the midpoints of the first and third intercostal spaces (open square) just lateral to the sternum as measured on the skin surface.

**Figure 2 jcm-11-03692-f002:**
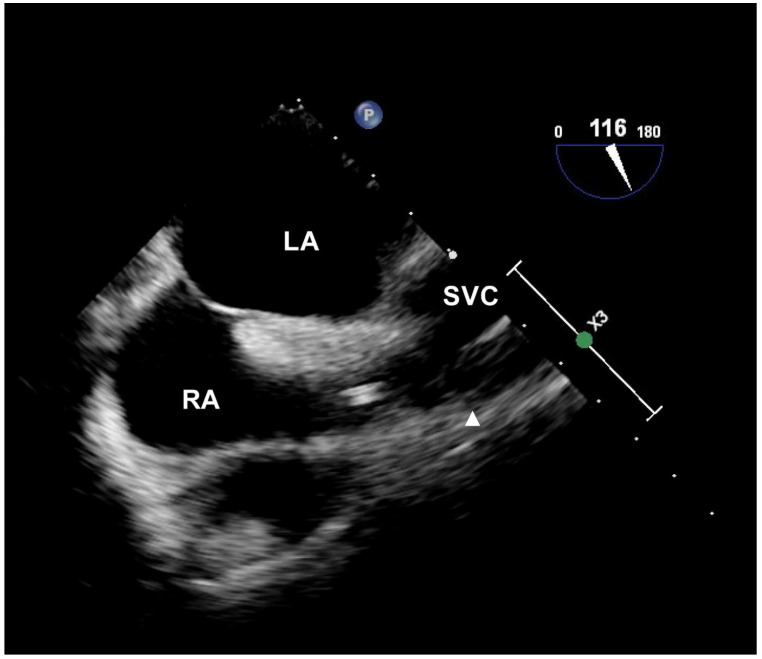
Echocardiographic image of catheter tip positioning. The catheter tip is identified as two parallel echogenic lines from the bicaval view. The solid triangle indicates the level of the upper border of the crista terminalis, defined as the echocardiographic junction of the SVC and the RA. Abbreviations: LA, left atrium; RA, right atrium; SVC, superior vena cava.

**Figure 3 jcm-11-03692-f003:**
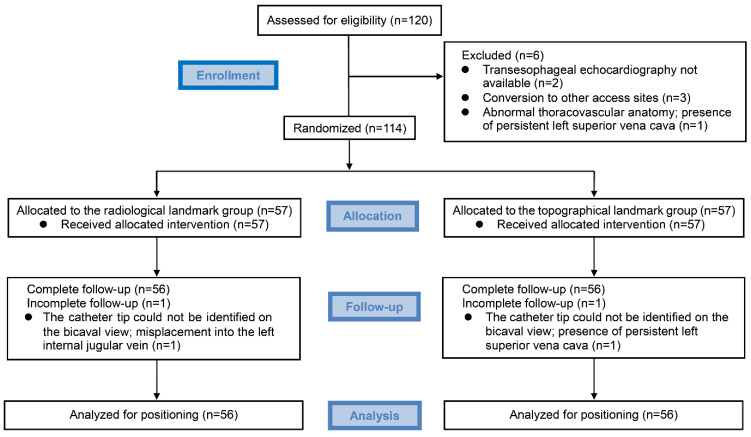
Consolidated Standards of Reporting Trials (CONSORT) flow diagram for participants included in the study.

**Figure 4 jcm-11-03692-f004:**
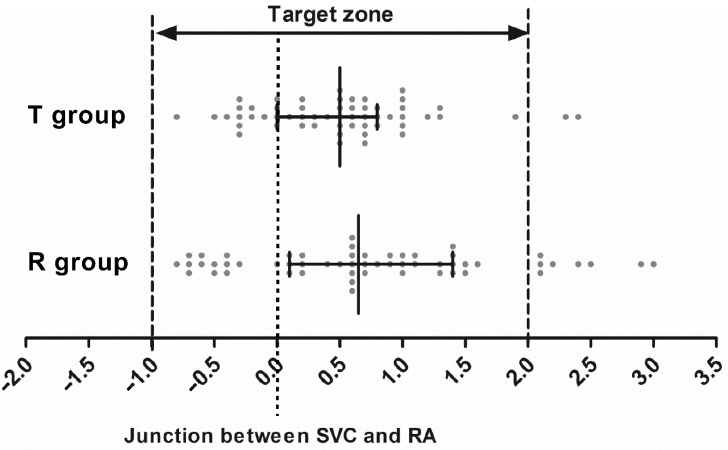
Scatter graph of catheter tip position within the target zone in both groups. Each circle represents an individual catheter tip position. Zero point refers to the junction between the SVC and the RA. Positive values indicate catheter tip position above the junction, and negative values indicate catheter tip position below the junction. Dashed lines indicate the upper and lower borders of the target zone. The solid vertical line indicates the median and the error bars indicate the interquartile range. Abbreviations: RA, right atrium; SVC, superior vena cava.

**Table 1 jcm-11-03692-t001:** Baseline characteristics of study participants.

Variable	Radiological Group (*n* = 56)	Topographical Group (*n* = 56)	*p*
Male sex	36 (64.3)	39 (69.6)	0.547
Age (years)	65.0 [58.0–71.8]	67.0 [59.3–75.0]	0.230
Height (cm)	163.0 [153.3–166.0]	164.0 [158.0–169.0]	0.079
Weight (kg)	64.0 [55.0–71.8]	63.0 [56.0–69.8]	0.818
BMI (kg/m^2^)	24.3 [21.9–26.5]	23.6 [21.7–26.2]	0.317

Values are reported as the median [interquartile range], number, or number (% of patients). Abbreviations: BMI, body mass index.

**Table 2 jcm-11-03692-t002:** Measurement and assessment in catheter positioning.

Variable	Radiological Group (*n* = 56)	Topographical Group (*n* = 56)	*p*
Catheter insertion depth (cm)	19.5 [18.6–20.4]	19.8 [18.8–20.2]	0.645
Actual distance to junction (cm)	20.5 [19.6–21.0]	20.4 [19.5–21.0]	0.802
Difference between measurements (cm)	0.7 [0.1–1.4]	0.5 [0–0.8]	0.171
Acceptable positioning	48 (85.7)	54 (96.4)	0.047 *
Position above target zone	8 (14.3)	2 (3.6)	0.047 *
Position below target zone	0	0	
Angle of tip (>40°) to the SVC	0	0	
Abutment with the SVC	1 (1.8)	0	0.315
Flow streams hitting vascular wall	1 (1.8)	1 (1.8)	1.000

Values are reported as the median [interquartile range], number, or number (% of patients). * Statistically significant differences between groups. Abbreviations: SVC, superior vena cava.

## Data Availability

The datasets used and/or analyzed during the current study are available from the corresponding author upon a reasonable request.

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
