# Peer review of "Accuracy of Catheter Positioning during Left Subclavian Venous Access: A Randomized Comparison between Radiological and Topographical Landmarks"

_jcm, 2022, doi:10.3390/jcm11133692_

Round 1
Reviewer 1 Report
General Comments
In their paper entitled “Accuracy of catheter positioning during left subclavian venous 2 access: a randomized comparison between radiological and 3 topographical landmarks”, authors
It is prospective randomized study comparing two methods for calculating the length of a left subclavian central venous catheter (CVC) a radiological and a topographical. The calculations were compared to a measurement using transesophageal ultrasound. Both groups contained 56 patients. It is an interesting well-designed paper, providing a useful landmark method for calculating CVC length.
Specific Comments
1) Abstract, P. 1, line 19: Please change “...the needle insertion point...” to “...the venous insertion point...” to avoid confusion with the needle insertion point in the skin.
2) Use this phrasing consistently, throughout the text, for example again in p. 4, line 130 “...distance between the needle insertion point...” it is not clear if you mean the skin insertion or the venous insertion point. Please rephrase.
3) Introduction, P. 2, line 35-39: Please rephrase the whole first sentence for more clarity, to something like “In a left subclavian venous access, the catheter tip may be positioned in the middle portion of the innominate vein to ensure a parallel course and to prevent SVC abutment. However, in this proximal position, it is prone to thrombosis due to the relatively small venous diameter, malfunction of the catheter owing to extravasation of the proximal access site, and infection in case of repositioning.”
4) P. 3, line 85: Please further specify the confusing term “...chest radiography...” to “...standard erect P-A chest radiograph in suspended full inspiration...”.
5) P. 4, line 129: Please change “...at the determined insertion depth...” to “...at the radiographically or topographically predetermined insertion depth...” for more clarity.
6) P. 4, line 132: “Following repositioning of the catheter tip at the right atrial junction...”. Was the use of a 20 cm catheter enough to reach the right atrial junction from the left subclavian vein in all patients? This seems unlikely, at least with taller individuals. I would suggest adding “...at the right atrial junction or at the maximum depth of the 20 cm catheter.”
7) P. 4, line 134: “Postoperatively, the catheter position was evaluated using chest radiography”. Do you mean recumbent or erect chest radiography in full inspiration, because the tip of the catheter may change position using different radiographic technique? Was the postoperative radiograph made using the same positioning as the preoperative radiograph used for measurement in the R-group? Please clarify.
8) P. 5, line 151: “Based on the landmarks and values of right-sided cannulation [...] patients were separated into two groups with the carina and the right third intercostal space as landmarks for positioning the catheter tip within the target zone.” Please rephrase completely for clarity. For example change “...in a previous study...” to “...from a...” “...into two groups...” to “...in...”, “...values of...” to “...calculated values for...”, etc.
9) P. 8, line 252: Please change “...innominate...” to “...brachiocephalic...”.
10) P. 9, line 277: “...from the axillary vein.” Please clarify if this patient was excluded from the cohort, in what group did the patient belong to and what was the reason for exclusion?
11) The comparison of a calculation made on a standard chest x-ray made with the patient erect and on a full inspiration to the measurement using transesophageal ultrasound with the patient supine and breathing should be mentioned on the limitations of the study, as the two positions are not truly identical.
12) Another limitation of the method worth mentioning would be that a confirmatory chest x-ray is still needed regardless of the calculation method used for CVC tip positioning.
13) As a limitation we could also add that the patients were subjected to transesophageal ultrasound with no clear benefit to them.
Author Response
1) Abstract, P. 1, line 19: Please change “...the needle insertion point...” to “...the venous insertion point...” to avoid confusion with the needle insertion point in the skin.
Response:
Thank you for this thoughtful comment. We agree that our description was insufficient. As per your suggestions, we have changed the phrase “the needle insertion point...” to “...the venous insertion point” in the revised text.
2) Use this phrasing consistently, throughout the text, for example again in p. 4, line 130 “...distance between the needle insertion point...” it is not clear if you mean the skin insertion or the venous insertion point. Please rephrase.
Response:
Thank you for this helpful comment. Based on your comment, we have consistently revised the phrase “the needle insertion point...” to “...the venous insertion point” throughout the text.
3) Introduction, P. 2, line 35-39: Please rephrase the whole first sentence for more clarity, to something like “In a left subclavian venous access, the catheter tip may be positioned in the middle portion of the innominate vein to ensure a parallel course and to prevent SVC abutment. However, in this proximal position, it is prone to thrombosis due to the relatively small venous diameter, malfunction of the catheter owing to extravasation of the proximal access site, and infection in case of repositioning.”
Response:
Thank you for this thoughtful comment. We have revised the Introduction section, P. 2, lines 36-38 accordingly.
4) P. 3, line 85: Please further specify the confusing term “...chest radiography...” to “...standard erect P-A chest radiograph in suspended full inspiration...”.
Response:
Thank you for this thoughtful comment. We agree that our description was insufficient. As per your suggestion, we have changed the phrase “...chest radiography...” to “...standard erect P-A chest radiograph in suspended full inspiration...” in the revised text.
5) P. 4, line 129: Please change “...at the determined insertion depth...” to “...at the radiographically or topographically predetermined insertion depth...” for more clarity.
Response:
Thank you for this thoughtful comment. We agree that our description was insufficient. According to your suggestion, we have changed the phrase “...at the determined insertion depth...” to “...at the radiographically or topographically predetermined insertion depth...” in the revised text.
6) P. 4, line 132: “Following repositioning of the catheter tip at the right atrial junction...”. Was the use of a 20 cm catheter enough to reach the right atrial junction from the left subclavian vein in all patients? This seems unlikely, at least with taller individuals. I would suggest adding “...at the right atrial junction or at the maximum depth of the 20 cm catheter.”
Response:
Thank you for this thoughtful comment. We agree that our description was insufficient. As you have accurately pointed out, a 20 cm long catheter did not reach the right atrial junction in all patients. In cases of a commercial length with a depth of 20 cm, we did not use the fixation hub.
We have changed the phrase “... at the right atrial junction...” to “...at the right atrial junction or at the maximum depth of the 20 cm catheter” in the revised text.
7) P. 4, line 134: “Postoperatively, the catheter position was evaluated using chest radiography”. Do you mean recumbent or erect chest radiography in full inspiration, because the tip of the catheter may change position using different radiographic technique? Was the postoperative radiograph made using the same positioning as the preoperative radiograph used for measurement in the R-group? Please clarify.
Response:
Thank you for this thoughtful comment. As you have accurately pointed out, the position of the catheter tip may change according to different radiographic techniques, patient’s position, and degree of inspiration. We did not use the same positioning in the postoperative radiograph compared to the preoperative radiograph since most of patients had undergone cardiothoracic and abdominal surgeries and could hardly maintain an erect position with full inspiration owing to immobility and pain. We evaluated the catheter migrations, aberrant position, and any other complications, such as hemothorax and pneumothorax, postoperatively.
We have revised the manuscript as follows;
“Postoperatively, the catheter position was rechecked using a recumbent chest radiograph upon inspiration at the bedside.”
8) P. 5, line 151: “Based on the landmarks and values of right-sided cannulation [...] patients were separated into two groups with the carina and the right third intercostal space as landmarks for positioning the catheter tip within the target zone.” Please rephrase completely for clarity. For example change “...in a previous study...” to “...from a...” “...into two groups...” to “...in...”, “...values of...” to “...calculated values for...”, etc.
Response:
Thank you for this thoughtful comment. We agree that our description was insufficient. As you pointed out, we have revised the manuscript as follows;
“Based on the landmarks and calculated values for right-sided cannulation from a previous study, the sample size was calculated from the data based on our preliminary observation, in which the patients were divided into two groups with the carina and the right third intercostal space as landmarks for positioning the catheter tip within the target zone.”
9) P. 8, line 252: Please change “...innominate...” to “...brachiocephalic...”.
Response:
Thank you for this thoughtful comment. We agree that our description was insufficient. As per your suggestion, we have changed the phrase “...innominate...” to “...brachiocephalic...” in the revised text.
10) P. 9, line 277: “...from the axillary vein.” Please clarify if this patient was excluded from the cohort, in what group did the patient belong to and what was the reason for exclusion?
Response:
Thank you for this thoughtful comment. We agree that our description was insufficient. In two patients who underwent the left subclavian venous access in this study, the catheter tips could not be identified on the bicaval view. These cases had incomplete follow-up and were excluded from the statistical analysis. According to your suggestion, we have revised the manuscript as follows;
“On the bicaval view of echocardiography, the catheter tip location could not be confirmed in two patients who were excluded from the statistical analysis owing to incomplete follow-up. These events occurred owing to the misplacement into the internal jugular vein and the presence of a persistent left SVC. Although misplacement is less common in left-sided subclavian venous access owing to an obtuse angle of the innominate vein with the SVC, it did occur in one participant in the R group [24]. It may be associated with a tortuous path and a more distal approach from the axillary vein.”
11) The comparison of a calculation made on a standard chest x-ray made with the patient erect and on a full inspiration to the measurement using transesophageal ultrasound with the patient supine and breathing should be mentioned on the limitations of the study, as the two positions are not truly identical.
Response:
Thank you for this thoughtful comment. We greatly appreciate your valuable suggestion. The patient position and breathing pattern applied for the calculation using a standard chest radiograph should be compared with that for measurement using transesophageal ultrasound since the positions and breathing patterns were not identical. As the erect chest radiograph was a routine screening test for preoperative evaluation in our hospital and transesophageal ultrasound was applied to patients who underwent general anesthesia and mechanical ventilation, it was not possible to control and maintain the same position and breathing pattern. According to your suggestion, we have added this comment to the limitations of the study and have revised the manuscript as follows;
“Third, we did not consider the difference in patient position and breathing pattern, wherein for the calculation using a preoperative standard P-A chest radiograph, the patient was in an upright position in full inspiration, while for the measurement using transesophageal ultrasound, the patient was in a supine position and mechanical ventilated.”
12) Another limitation of the method worth mentioning would be that a confirmatory chest x-ray is still needed regardless of the calculation method used for CVC tip positioning.
Response:
Thank you for this thoughtful comment. In addition to this comment, the authors were also inspired by the 13th comment. We greatly appreciate your valuable suggestion. According to your suggestion, we added this comment to the text and have revised the manuscript as follows;
“These patients underwent transesophageal ultrasound for the measurement of catheter tip positioning; however, no clear benefit was observed. A confirmatory chest radiograph is still needed regardless of the calculation method used for catheter tip positioning.”
13) As a limitation we could also add that the patients were subjected to transesophageal ultrasound with no clear benefit to them.
Response:
Thank you for this thoughtful comment. These points are described in the text and we have revised the manuscript as follows;
“These patients underwent transesophageal ultrasound for the measurement of catheter tip positioning; however, but no clear benefit was observed. A confirmatory chest radiograph is still needed regardless of the calculation method used for catheter tip positioning.”

Reviewer 2 Report
Dear Authors, I thank you for the paper submitted. I think it is very interesting. However I have some comments.
First, I think that the difference of tip positioning between groups was clinically insignificant (although statistically significant). Please, emphasise this concept.
Second, specify how was done chest Xray. I mean: was the patients supine? semirecumbent? Upright? I think it is important to clarify this.
Third, I have some concerns regarding the sample size you calculated. Based on your assumptions, I estimate at least 150 patients to be enrolled to satisfy your question. Please, clarify.
Best regards
Author Response
First, I think that the difference of tip positioning between groups was clinically insignificant (although statistically significant). Please, emphasise this concept.
Response:
Thank you for this thoughtful comment. We sincerely appreciate your valuable suggestion.
The placement of the central venous catheter is always associated with risks, and the optimal positioning of the catheter tip is an ongoing controversial issue, especially considering left-sided venous access.
Vascular injuries are possible in any position in the SVC and cardiac chamber. Most devastating complications, such as cardiac tamponade, hemothorax, and hydrothorax, were reported in case reports and were attributed to mechanical and chemical irritation to the vascular wall, which were related to parenteral delivery of hyperosmolar solutions, migration of the tip from the arm movement in a peripherally inserted central venous catheter, rigid catheter material, and acute angle from left-sided access.
The most important points to be considered for preventing these events are the alignment of the catheter tip to the vessel wall, free movement of the tip, and non-impingement to the vessel wall. The distal SVC close to the RA/SVC junction has the advantage of ensuring a parallel pathway for the catheter tip and large conduit during cardiac pulsation.
Several methods and landmarks are used for catheter positioning in right-sided vascular access. However, there is no definite landmark in left-sided access for catheter tip positioning in the distal SVC close to the RA/SVC junction. Therefore, we evaluated catheter tip positioning at the distal SVC close to the RA/SVC junction using recognized anatomical landmarks.
In our study, the catheter tip was more accurately positioned in the distal SVC close to the RA/SVC junction using topographical landmarks than radiological landmarks (96.4% vs. 85.7%, respectively, P = 0.047). Although both the methods demonstrated comparable outcomes for the determined insertion depth (19.5 [18.6–20.4] cm vs. 19.8 [18.8–20.2] cm, respectively; P = 0.645), the actual distance to the right atrial junction (20.5 [19.6–21.0] cm vs. 20.4 [19.5–21.0] cm, respectively; P = 0.802), and the difference between the measurements (0.7 [0.1–1.4] cm vs. 0.5 [0–0.8] cm, respectively; P = 0.171), the radiological method had a higher incidence of the catheter tip being located 2 cm above the right atrial junction (14.3% vs. 3.6%, respectively, P = 0.047). Therefore, we recommend that the right first and third intercostal spaces should be used as landmarks for positioning the catheter tip close to the right atrial junction during left subclavian vein cannulation. If identifying the surface landmarks is challenging, radiological landmarks, such as the carina, can alternatively be used for positioning the catheter.
According to your suggestion, the emphasis on clinical importance has been described in the discussion section, and the manuscript has been revised as follows;
“The placement of the central venous catheter is always associated with risks, and the optimal positioning of the catheter tip is an ongoing issue, especially in left-sided venous access. Vascular injuries are possible in any position in the SVC and cardiac chamber. Most devastating complications, such as cardiac tamponade, hemothorax, and hydrothorax, were reported in case reports and were attributed to mechanical and chemical irritation to the vascular wall, which were related to parenteral delivery of hyperosmolar solutions and acute angle from left-sided access. The most important points to be considered for preventing these events are the alignment of the catheter tip to the vessel wall, free movement of the tip, and non-impingement to the vessel wall. The distal SVC close to the right atrial junction has the advantage of ensuring a parallel pathway for the catheter tip and large conduit during cardiac pulsation.
Several methods and landmarks are used for catheter positioning in right-sided vascular access. However, there is no definite landmark in left-sided access for catheter tip positioning in the distal SVC close to the right atrial junction.
In our study, although both the methods demonstrated comparable outcomes, the radiological method had a higher incidence of the catheter tip being located 2 cm above the right atrial junction. The catheter tip was more accurately positioned in the distal SVC close to the right atrial junction using topographical landmarks than radiological landmarks.”
Second, specify how was done chest Xray. I mean: was the patients supine? semirecumbent? Upright? I think it is important to clarify this.
Response:
Thank you for this thoughtful comment. We sincerely appreciate your valuable suggestion. The patient position and breathing pattern applied for the calculation using a standard chest radiograph should be compared with that for measurement using transesophageal ultrasound since the positions and breathing patterns were not identical. As the erect chest radiograph was a routine screening test for preoperative evaluation in our hospital and transesophageal ultrasound was applied to patients who underwent general anesthesia and mechanical ventilation, it was not possible to control and maintain the same position and breathing pattern. According to your suggestion, we have added this comment to the limitations of the study and have revised the manuscript as follows;
“Third, we did not consider the difference in the patient’s position and breathing pattern wherein for the measurement using a preoperative standard P-A chest radiograph, the patient was in an upright position in full inspiration, while for the measurement using transesophageal ultrasound, the patient was in a supine position and mechanical ventilated.”
Third, I have some concerns regarding the sample size you calculated. Based on your assumptions, I estimate at least 150 patients to be enrolled to satisfy your question. Please, clarify.
Response:
Thank you for this thoughtful comment.
As described in the main manuscript, the estimation of the sample size was performed based on the results obtained by a pilot study using PASS 12 (NCSS, LLC, Kaysville, UT, USA) of the Radiological group 83.3% (25/30) and Topographic group 96.6% (29/30). The execution of PASS 12 is shown in the figure of this response letter. An alternative one-sided hypothesis was selected, stating that P2 (Topographic group: 96.6%) was greater than P1 (Radiological group: 83.3%), and the pooled Z test was selected for the test type. In addition, for the RCT study, the same number of sample sizes was obtained, with the ratio of each group being 1. Power and alpha were selected as 0.80 and 0.05, respectively. The result was calculated as 56 cases each of N1 and N2 at power 0.8026, as shown in the figure.
Thank you for the good point. We tried to clarify the processes of obtaining the sample size by describing these conditions in the revised manuscript. In the revised manuscript, we have mentioned the one-side hypothesis and pooled Z test.
In the methods section, the authors described and inserted the relevant text to address the reviewer’s concern as follows;
“Based on the landmarks and calculated values for right-sided cannulation from a previous study, the sample size was calculated from the data based on our preliminary observation, in which the patients were divided into two groups with the carina and the right third intercostal space as landmarks for positioning the catheter tip within the target zone [8]. Consequently, 30 patients were included in each group, and the incidence of acceptable positioning was 83.3% (25/30) and 96.6% (29/30) in the carina and the third intercostal space groups, respectively. Based on the incidence rate, an alternative hypothesis and test type were chosen as one-sided (H1: P1<P2) and pooled Z test, respectively. We calculated that 56 patients were required in each group to detect a difference of this magnitude with an α error of 0.05 and desired power of 0.80, using PASS 12 (NCSS, LLC, Kaysville, UT, USA). After accounting for a dropout rate of 6%, we recruited 120 patients for this study.”
Figure 1. Method selection screen: The method surrounded by a red box was used. Tests for Two Proportions using Proportions
Figure 2. Condition entry process
Figure 3. Result screen: There are sample sizes and power selected in the red box.

Round 2
Reviewer 2 Report
Dear Authors, you answered my issues.
Best regards